# Data integration between clinical research and patient care: A framework for context-depending data sharing and in silico predictions

Katja Hoffmann[1,2]*, Anne Pelz[1], Elena Karg[1], Andrea Gottschalk[1], Thomas Zerjatke[1], Silvio Schuster[1], Heiko Böhme[2], Ingmar Glauche[1], Ingo Roeder[1,2]

1 Institute for Medical Informatics and Biometry, Faculty of Medicine Carl Gustav Carus, Technische Universität Dresden, Dresden, Germany, 2 National Center for Tumor Diseases (NCT/UCC), Dresden, Germany: German Cancer Research Center (DKFZ), Heidelberg, Germany; Faculty of Medicine and University Hospital Carl Gustav Carus, Technische Universität Dresden, Dresden, Germany; Helmholtz-Zentrum Dresden - Rossendorf (HZDR), Dresden, Germany

* katja.hoffmann@tu-dresden.de

## Abstract

The transfer of new insights from basic or clinical research into clinical routine is usually a lengthy and time-consuming process. Conversely, there are still many barriers to directly provide and use routine data in the context of basic and clinical research. In particular, no coherent software solution is available that allows a convenient and immediate bidirectional transfer of data between concrete treatment contexts and research settings. Here, we present a generic framework that integrates health data (e.g., clinical, molecular) and computational analytics (e.g., model predictions, statistical evaluations, visualizations) into a clinical software solution which simultaneously supports both patient-specific healthcare decisions and research efforts, while also adhering to the requirements for data protection and data quality. Specifically, our work is based on a recently established generic data management concept, for which we designed and implemented a web-based software framework that integrates data analysis, visualization as well as computer simulation and model prediction with audit trail functionality and a regulation-compliant pseudonymization service. Within the front-end application, we established two tailored views: a *clinical (i.e., treatment context) perspective* focusing on patient-specific data visualization, analysis and outcome prediction and a *research perspective* focusing on the exploration of pseudonymized data. We illustrate the application of our generic framework by two use-cases from the field of haematology/oncology. Our implementation demonstrates the feasibility of an integrated generation and backward propagation of data analysis results and model predictions at an individual patient level into clinical decision-making processes while enabling seamless integration into a clinical information system or an electronic health record.

**Data Availability Statement:** The source code of the latest server application can be downloaded from the GitLab repository at https://gitlab.com/imb-dev/predictdemo. Furthermore, the source code archived at the time of publication can be found at https://zenodo.org/record/7655167#.Y_Kdiy1XaqA. This repository also includes the computational models and test datasets implemented in the demo server as well as the developer documentation, including initial installation instructions.

**Funding:** This work was supported by the German Federal Ministry of Education and Research (BMBF) project "prediCt" in the framework of the ERA-Net ERACoSysMed JTC-2 (BMBF grant number 031L0136A to IR). The funders had no role in study design, data collection and analysis, decision to publish, or preparation of the manuscript.

**Competing interests:** The authors have declared that no competing interests exist.

## Author summary

Patient-oriented research is based on comprehensive, quality-assured medical data that is visualized and analysed to gain knowledge. Based hereon, computer models can be developed, which e.g., calculate risk scores or predict treatment success. Such approaches can be used for risk staging or for selecting the optimal therapy for a specific patient. In recent years, a lot of efforts have been made to develop generic concepts for data processing and for providing the data in the research context. What has been missing so far is a suitable software infrastructure to facilitate the direct backward propagation of scientific results into everyday clinical practice to support the treating clinicians in their decision-making processes. To close this gap, we designed a generic software framework into which, in principle, any computational model or algorithm can be integrated. For demonstration purposes, we developed a web application that integrates two mathematical models from the field of haematology, specifically relating to chronic myeloid leukaemia (CML). Both models calculate the leukaemia recurrence probability of a specific patient, after the intended stopping of the applied therapy. The particular prediction is based on patient-specific molecular diagnostic data and can be used for personalized treatment adaptation.

## Introduction

Mathematical models and computational algorithms have already proven useful in analysing data, explaining functional relationships of (patho-)physiological mechanisms, and guiding clinical research. Along this line, also computational applications ("apps") are increasingly used to support clinical decision-making [1–3]. Prominent examples are risk scores, which provide patient-specific risk estimates by integrating relevant clinical parameters based on mathematical/statistical models [4–6]. Furthermore, mathematical models and computer simulations for disease progression and treatment response have been developed for several diseases [7–14]. As these models are intrinsically based on clinical data, there is an increasing need to provide such data in a structured and secure way. To fully realize the translational potential of both, models and data, it is necessary to establish interfaces that allow the integration of clinical data and corresponding model predictions and provide them in a hospital setting.

In our previous work [15] we focused on how patient-specific model predictions can be provided to clinicians within a particular clinical information system (CIS), while we have not considered the full life cycle of clinical and simulation data. However, due to privacy and security aspects, the question arises how health data from multiple, decentralized data sources can be consistently integrated and made available for computer simulations and analytics in general and how analytic and simulation results, e.g., generated to predict an individual patient's future behaviour, can be transferred back to the routine clinical practice. Currently, we are missing technically applicable concepts to integrate the results of data analyses and mathematical model predictions in a hospital setting to support decision-making at an individual patient level. Therefore, the aim of this work is to develop a generic concept for context-depending data-sharing and integration of computer simulations for both scientific research and routine care in a common hospital setting that goes beyond our previously published framework [15], specifically in terms of centralised and privacy-compliant management of patient-identifying and medical data and thus for simultaneous use in clinical and research settings. In addition, a software prototype has been developed that exemplifies the functionality of the framework based on possible use cases.

## Materials and methods

### Data, process and requirement analysis

Based on the conceptional works on data integration in scientific research [16], we investigated how quality-control, data protection, and ethical requirements can be guaranteed when integrating medical data in the research and clinical context in parallel. This was done in close collaboration with the Data Integration Centre of the University Hospital Dresden, Germany and the Independent Trusted Third Party of the TU Dresden, Germany. We took advantage of the fact that hospitals currently implement data management concepts for the efficient and legally compliant provision and use of health data with permanent traceability of all data sources with a clear origin, time stamp and authorship. For this purpose, many hospitals are establishing Research Data Management Systems for data pre-processing, storage and provision, sometimes coupled with an Independent Trusted Third Party (TTP) to centrally regulate personal identifying data and to manage consents, revocations, and pseudonym assignments. Fig 1

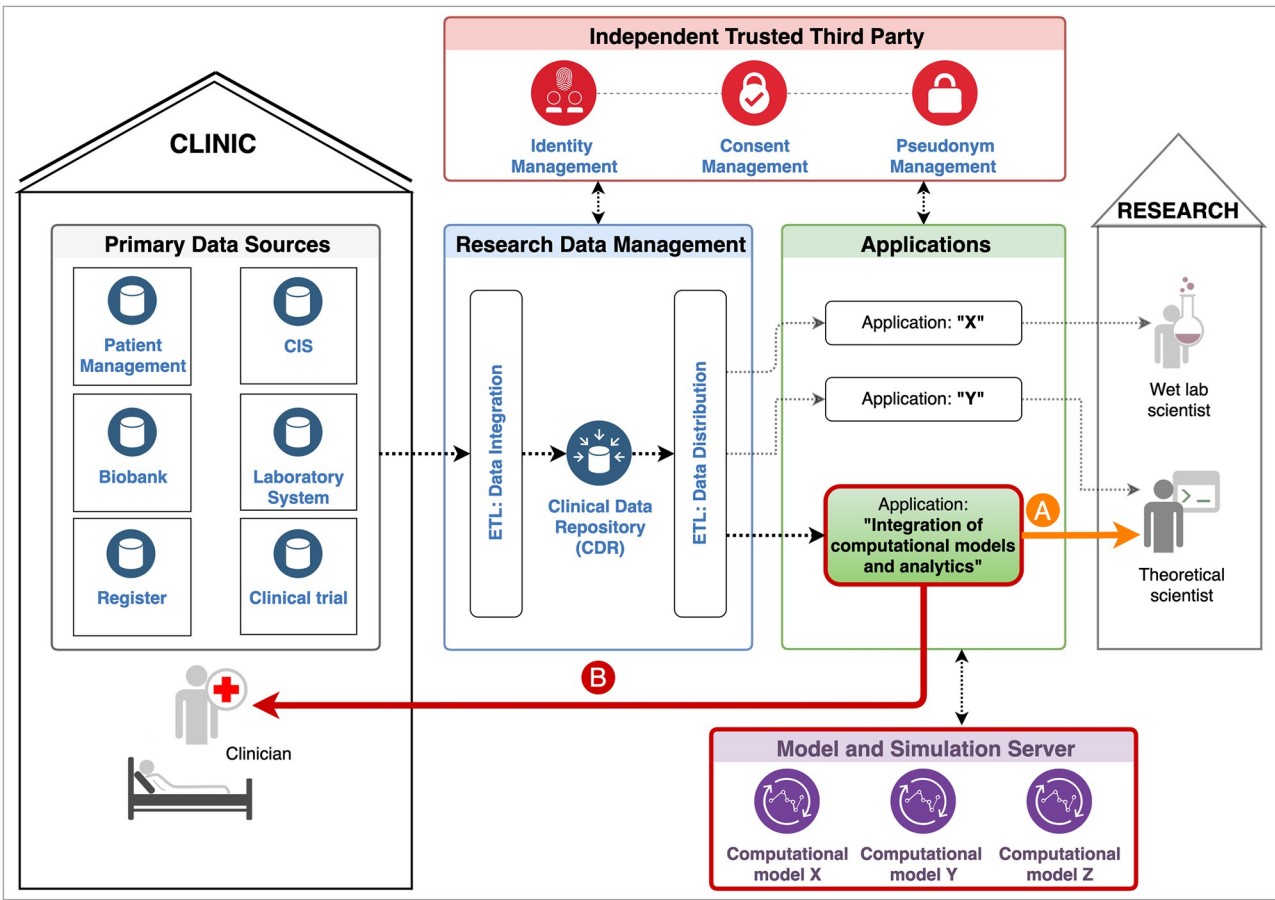

**Fig 1. Example of a generic configuration of data management from the primary data sources to the applications.** Patient data from various, decentralized, and heterogeneous data sources (Primary Data Sources) are extracted, transformed to harmonized data and loaded into a Clinical Data Repository (Extract-Transform-Load (ETL): Data Integration). Data transfer points pseudonymize these data and make them available for analysis and specific applications (ETL: Data Distribution). The identifying data, consents, and the pseudonym linkages are managed by an Independent Trusted Third Party or another pseudonymised service/provider. This institution/service also verifies the existence of the required consent before any processing of personal data. Our specific application "Integration of computational models and analytics", which is additionally connected to a Model and Simulation Server, is intended to be used in clinical research (A) and also in clinical practice (B). Both components (outlined in bold red) complement the existing concept of data management in scientific research.

shows an example of a generic configuration of data management components from the primary data sources to the applications in patient-oriented research.

Such a typical configuration comprises the selective extraction of heterogeneous patient data, which are stored in various primary data sources, the transformation into a unique target structure, and the loading into a Clinical Data Repository (CDR) (cf. Fig 1 "ETL: Data Integration"). Such data processing is often denoted as ETL (**E**xtract-**T**ransform-**L**oad) process. In order to guarantee a highest level of data quality, appropriate methods (e.g., data audit, standardization, data cleansing) are integrated within the "ETL: Data Integration" step (see Fig 1) to clean and homogenize the source data before importing it into the CDR. The TTP verifies the existence of informed consent for the usage of the required data and provides application-specific pseudonyms. This ensures that the pseudonymized medical data do not allow any conclusions about a patient's identity. In the case of human genetic data or data on rare diseases, where re-identification might be possible in principle, specific actions are taken to comply with data protection regulations. Eventually, data transfer points (cf. Fig 1 "ETL: Data Distribution") provide the pseudonymized medical data requested for specific applications, which can be used by authorized researchers (e. g. clinical scientists, data analysts, modelers).

In contrast to other tools and workflows that have been already developed to provide health data for research (e.g. [17,18]), illustrated as the "A" arm in Fig 1, the main focus of our approach is on the backward propagation of research findings and data analysis results into the clinical process, illustrated as the "B" arm in Fig 1. Specifically, we investigated how patient-specific predictions based on mathematical models and corresponding computer simulations can be integrated directly into the clinical workflows to support actual clinical decision-making for treatment optimizations.

Our work emerges from a previously established demonstrator software [15] that integrates time course data for chronic myeloid leukaemia (CML) patients under ongoing therapy to provide estimates of their future behaviour. While the focus on CML is a particular example application, the software prototype itself is flexible to integrate other types of data and underlying models. The application is primarily intended to be used by clinicians as a clinical decision support tool, complementary to existing primary data sources to optimize treatment strategies on a patient-specific level (see Fig 1B). In addition, it can also be used by researchers in the context of scientific projects (see Fig 1A). For both use cases, the software application has to fulfil the following criteria:

1. Collection, processing, and provision of quality-assured pseudonymized data compliant with the Data Protection Laws (European) General Data Protection Regulation (*Datenschutzgrundverordnung DSGVO/GDPR*) and the German Federal Data Protection Law (*Bundesdatenschutzgesetz BDSG*).

2. Continuous provision of mathematical model predictions and data analyses in case of changes in the underlying data (treatment, observation, diagnosis, etc.).

3. Permanent traceability of all data and computational results with a unique origin, time stamp, and authorship (i.e., audit trail functionality).

4. Interactive visualization of medical data and simulation results.

5. Assignment of individual clinical data and model predictions and/or analyses to individual patients in clinical practice (re-identification of pseudonymized data).

6. User and access control by role-based right management.

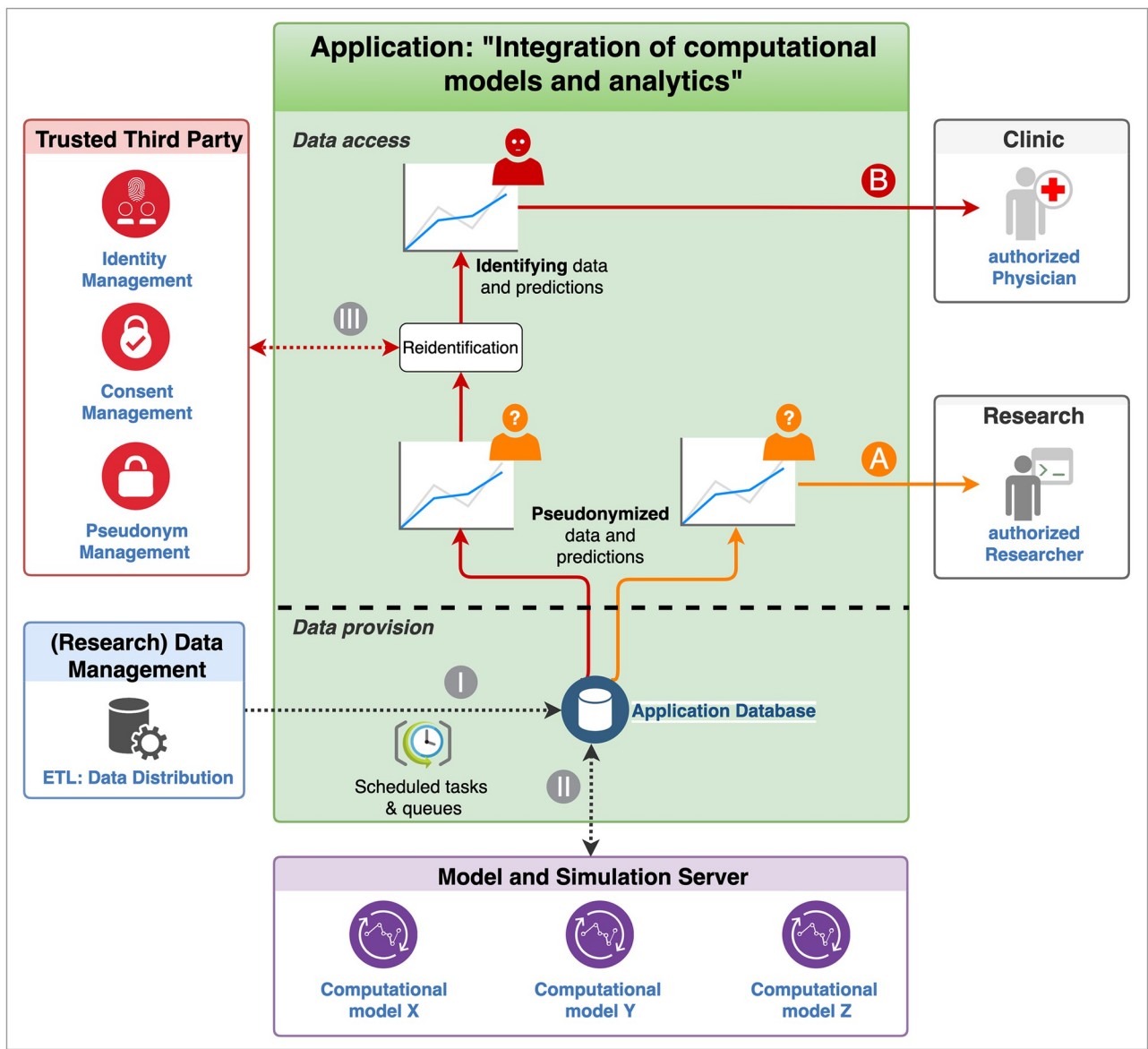

**Fig 2. Data provision and access of the generic software framework.** The Application Database contains exclusively pseudonymized medical data transferred by the data transfer point of the hospital's (Research) Data Management (I) and pseudonymized model predictions of particular computational models calculated by a model and simulation server for modelling and analysis (II) through scheduled tasks. Authorized researchers can retrieve the pseudonymized medical data and model predictions (A). For authorized clinicians, who have to be able to retrieve identifying medical data and model predictions (B) in clinical practice, a reidentification step (III) through the Trusted Third Party has been integrated.

In the following, we describe our generic software framework (see Fig 2) that ensures all the specified requirements and in which algorithmic results (e.g., model predictions, analytic results) can be made available to researchers and clinicians directly in their working environment in an accurate, privacy compliant and transparent way. We outline the newly developed web application (cf. Software prototype, the demo server as an instance of the software prototype at https://tu-dresden.de/med/demoserver and S1 File) to exemplify the generic software framework for a particular clinical application from the field of haematology/CML.

## Generic overall design and data flows

Based on the generic concept of data management in patient-oriented research (c.f. Fig 1), we developed a software framework that demonstrates how mathematical model predictions, and —in a broader context—any algorithmic/computational procedure can be integrated into clinical practice to support decision-making. Most importantly, our solution strictly complies with standards for data security and pseudonymization, by following the guideline [19] of the *Data Protection Working Group of the Technology, Methods and Infrastructure for Network Medical Research TMF e.V.* [20], and allows audit trail compatible reconstruction of all of the provided analysis results/predictions (i.e., full transparency). To demonstrate the functionality of the software framework, we use the newly developed web application (software prototype) as a particular computational application. It is complemented by a model and simulation server, a TTP/pseudonymization server, and an interface to the (Research) Data Management of the hospital. Fig 2 shows the overall design of the framework and illustrates the processes of data provision and access, which are described below.

Pseudonymized medical data, as input for any data analysis and calculation of model predictions, are provided by a (Research) Data Management (cf. Figs 1 and 2I). This can be a CDR or any database storing medical data from (various) primary data sources, or also the primary data source itself (cf. Fig 1). Only a suitable data distribution has to be established. Here, the (Research) Data Management transfers exclusively pseudonymized data. No patient-identifying data is stored in the Application Database. After the provision of new or updated medical data, new computer simulations are scheduled periodically or event-driven. The results are provided by the Model and Simulation Server and transferred to the Application Database (Fig 2II). In this way, we ensure that the Application Database contains up-to-date simulation results/model predictions based on the most recent pseudonymized medical data.

The data access is regulated by a role-based Access and Rights Management to establish different views on the data, e.g., trial-specific *roles* or department-specific *roles*. Members of the trial-specific roles can survey a whole spectrum of *pseudonymized* patients from one or more specific clinical trials, potentially involving different sites (see section Research view). Members of the department-specific role (e.g., haematology) have permission to access to the *full data* set of his/her patients treated in the particular centre (see section Clinic view). In the treatment context, the reidentification of pseudonymized medical and simulation data is performed by the TTP (Fig 2III).

## Software prototype

We used the high-level Python Web framework Django [21,22] to build the frontend web application. Django provides robust security features, a user authentication system, and a fully featured admin interface.

The computer simulations (e.g., model predictions) are generated using an instance of the model and simulation server MAGPIE, which is a software framework, designed for publishing and executing computational models. MAGPIE also offers a full reproducibility of the calculated model predictions (i.e., audit trail functionality). For parameterization and execution of computational models and simulations, the "MAGPIE-API-R" package [23] is used. For a detailed description of the MAGPIE framework and implementation, we refer to Baldow et al. 2017 [24].

The Python framework Plotly/Dash [25] is used to build the interactive visualizations of medical and simulation results, which are seamlessly embedded in the frontend application.

For identity, consent, and pseudonym management, which are accomplished by the TTP, the MOSAIC Toolbox [26], consisting of an identity management module "EPIX" (**E**nterprise

Patient Identifier Cross-referencing), a consent management module"gICS" (generic Informed Consent Service), a pseudonym management module "gPAS" (generic Pseudonym Administration Service) and the TTP dispatcher (TTP Workflow Manager Module) developed within the MOSAIC project [27] are used. The pseudonymized data is reidentified by accessing the TTP Dispatcher [18] using the representational state transfer protocol (REST).

The demo server is based on an Apache HTTP Server [28] and uses PostgreSQL [29] as a database backend.

## Computational models and analytics

In the software prototype (cf. Results), two mathematical models for predictions on a patient level and a statistical analysis on aggregated data (see below for details) have been implemented for demonstration purposes. The framework itself is not restricted to these two particular models. It allows the deployment of different mathematical models and other algorithms, as long as they are registered in the MAGPIE model database, and feeding generated model predictions into the described workflow. There is no general restriction, neither on the model type nor on the particular implementation/programming language.

**Example model 1: Molecular monitoring during dose reduction in CML patients.** The DESTINY trial (#NCT01804985) investigated whether a reduction of tyrosine kinase inhibitor (TKI) dose in CML patients prior to treatment stop can lead to better treatment-free remission rates compared to full dose treatment [30]. Based on a reanalysis of these molecular response data, we showed that an increasing BCR-ABL level during the dose reduction phase is indicative for CML recurrences after TKI stop and can be used to risk-stratify patients prior to treatment cessation [5]. Given the individual molecular monitoring data during dose reduction, the statistical model implemented within the software prototype calculates the local slope parameter of the BCR-ABL dynamics and provides a prediction of the patient-specific recurrence probability [5]. In order to correctly interpret the model predictions, it is advisable to provide a suitable confidence measure. As an example for our particular use case, we implemented the 95% confidence interval for the estimated recurrence probability as described in [5].

**Example model 2: Estimation of immunological leukaemia control.** We also developed a mathematical model of CML-immune interaction [31], which can be fitted to available time course data of TKI-treated CML patients and after therapy stop. We showed that qualitatively different dynamic treatment response patterns can be characterized in the context of a mathematical model by different stable steady states, describing the individual immune response [32]. The different steady states can be interpreted as "immune classes". Specifically, we distinguish between an insufficient immune response (class A) for which a TKI stop does not seem to be appropriate, a sufficient immune response for which one can expect leukaemia control after a TKI stop (class B), and a weak immune response (class C) for which the effectiveness of the immune response cannot be clearly predicted. For a given set of data (as provided within the software prototype), this model can be used to infer patient-specific immune parameters and, therefore, to estimate a probability for his/her assignment to the three immune classes. As the estimated immune parameter are correlated with the patients' recurrence behaviour after treatment cessation, this classification can be used to support the decision-making with respect to TKI stopping.

**Example statistics: Estimating median time courses.** To evaluate particular patient-specific disease dynamics e.g., in comparison with an overall response pattern (e.g., across different studies), one can aggregate data from patient groups or trials. As an example, we calculate and visualize the median and interquartile ranges of the quantitative molecular measurements at different time intervals during treatment (see [33]). This feature is based on pseudonymized

data and can be used also outside the treatment context (research view) for all data that are accessible for the particular user (cf. access control).

## Results

### Software prototype

To demonstrate the generic software framework (cf. Generic overall design and data flows), we:

a. integrated a specific implementation of a model and simulation server (termed MAGPIE [23]) to manage and execute computational models.

b. equipped MAGPIE with two mathematical models from the field of haematology/CML (cf. Computational models and analytics).

c. integrated the MOSAIC TTP server for identity, consent and pseudonym management.

d. added fictitious patients to the identity management module (E-PIX) of the MOSAIC TTP server for illustration purposes.

e. equipped the consent management module (gICS) of the MOSAIC TTP server with informed consents for various CML trials (Research view) and the data usage within the haematological treatment context (Clinic view).

f. generated pseudonyms for each informed consent form using the pseudonym management module (gPAS) of the MOSAIC TTP server.

g. designed ETL processes for transferring the medical data into the "Application database" that are needed for the specific analyses and model predictions.

h. implemented a role-based user and access management and established roles for the "Clinic view" and "Research view" presentation.

The demo server, as an instance of the software prototype, is available at https://tu-dresden.de/med/demoserver. A detailed description of the demo server can be found in the following sections and the demo server walkthrough video (S1 File). The source code of the latest server application can be downloaded from the GitLab repository at https://gitlab.com/imb-dev/predictdemo. Furthermore, the source code archived at the time of publication can be found at https://zenodo.org/record/7655167#.Y_Kdiy1XaqA. This repository also includes the computational models and test datasets implemented in the demo server as well as the developer documentation, including initial installation instructions.

### Access control

The software prototype offers a role-based user and access management and can be adjusted to reflect department-specific roles (e.g., to be accessible for the Haematology Department) and trial-specific roles (to be limited to units involved in a given clinical trial e.g., the "Demonstrator CML Test Trial"). Each role corresponds to a consent implemented in the gICS module of the MOSAIC TTP server. We emphasize at this point that data objects for consent and pseudonyms are generated not only for research trials but also for the respective treatment context/department in patient care, even if no signed consent is required for this use. This generic data concept ensures that only authorized users can access particular patient data for particular purposes. Depending on the specific role, the user interface menu provides access to patient-specific data within a treatment context (here within the haematological treatment context; see

section Clinic view) and/or to pseudonymized data of particular trials (see section Research view). For testing purposes, the software prototype provides guest accounts with the ability to switch between the department specific and the trial-specific roles.

## Clinic view

Depending on the user role and the corresponding consent of the gICS module of the MOSAIC TTP server, the user can search for identifying patient data via a particular search form. For testing and demonstration purposes, the identity management module of the MOSAIC TTP server is equipped with a selection of fictitious patients, inspired by but not identical to patient time courses from the DESTINY trial (NCT01804985, [34]).

Once a particular patient had been identified, the gPAS module of the MOSAIC TTP server provides the pseudonym, which is then used to retrieve the medical data and/or the computer simulation results from the Application Database (*re-identification*, see Fig 2III).

The "Clinic view" (Fig 3) presents the current medical data (D) and the latest model predictions (E) of the requested patient visualized in a dashboard (C1). The prediction of the recurrence probability after stopping the TKI treatment and an estimate of the confidence interval is obtained from the model "Molecular monitoring during dose reduction" (cf. Computational models and analytics) and further presented in a simplified and optionally in an expert view ($E_1$ and $E_2$). For a comprehensive description of the model, a DOI link to the publication is provided ($E_3$). The current medical data can be also retrieved and downloaded in form of a table view (Fig 3C2). Finally, the complete history of calculated computer simulation/model prediction can be followed (Figs 3C3 and 4), which allows an audit trail functionality of the computational results. Fig 4 shows a model-predicted immune classification based on the "Estimation of immunological leukaemia control" (cf. Computational models and analytics). For further details about the computational models see section Materials and methods and the implementation of the demo server at https://tu-dresden.de/med/demoserver.

To demonstrate how the Clinic View can be implemented in a CIS or an Electronic Health Record (EHR), the software prototype provides a REST API to query all model predictions for a patient within a treatment context using the local CIS/EHR identifier. As a result, the CIS/EHR system receives a list of model predictions with metadata, including a link to a view of model results that can be accessed through or embedded in the CIS/EHR system. A description, including instructions on how to test the API, can be found in the developer documentation included in the repository or the demo server documentation website.

## Research view

Controlled by the access management, the software prototype additionally offers the ability to access pseudonymized data (e.g., from clinical trials) and corresponding, aggregated analyses and simulation/model results (see Fig 5). The demo server yields clinical datasets from several trials from the field of haematology, specifically recent CML trails. Depending on the particular permission status, the user can or cannot access the detailed information of a given clinical trial.

The "Detail View" of a specific clinical trial, here the "Demonstrator CML Test Trial" (see Fig 6), provides general information (A), the ability to access and download the pseudonymized medical data (B) and finally, the summary statistics and graphical illustrations for aggregated data (C). Further information about the analyses is provided in the section Example statistics. We wish to emphasize that the specific data analytics serve as an example to demonstrate the general ability of the framework, which can be flexibly extended or adapted to other further/different computational analytics and graphical representations according to the needs of the physician and/or researcher.

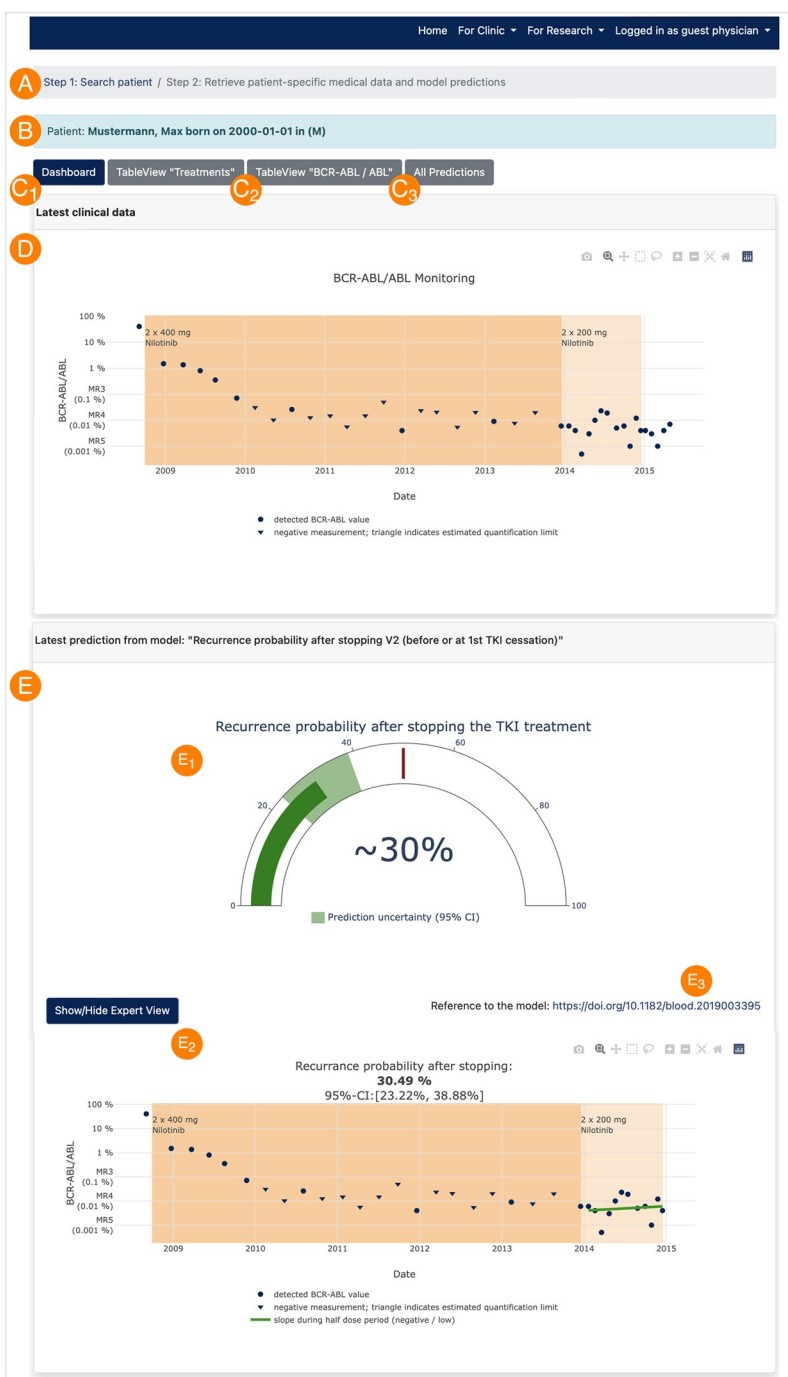

**Fig 3. Clinic view.** After searching for a patient by identifying data (A) and re-identification (B) of the pseudonymized medical data and simulation results, the latest medical data (D) and model predictions (E) are visualized as simplified and/or expert view ($E_1$ and $E_2$) in a dashboard (C1). For a comprehensive description of the model, a DOI link to the publication is provided ($E_3$). Furthermore, the current medical data can be displayed and downloaded in table form (C2) and all previous predictions can be retrieved, too (audit trail compatibility, (C3) and Fig 4).

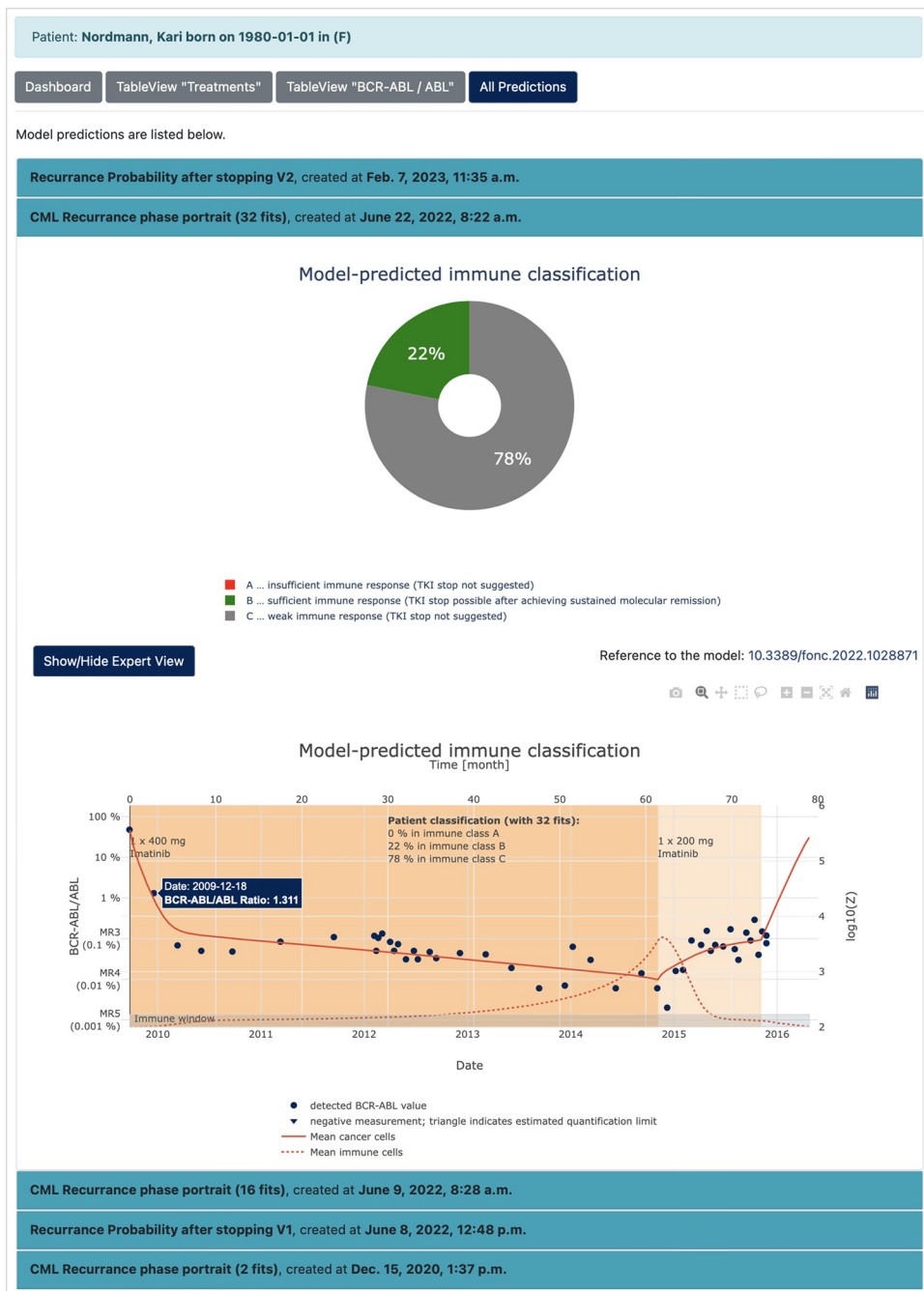

**Fig 4. Audit trail functionality.** Availability of every executed computer simulation/model prediction.

## Discussion

We have investigated how data analytics, specifically mathematical model predictions and computer simulation results can be integrated into a clinical routine environment to support clinical decision-making at an individual patient level and, simultaneously, support scientific research at the level of pseudonymized data. In particular, we examined how quality-assured and harmonized health data from multiple decentralized clinical data sources can be provided

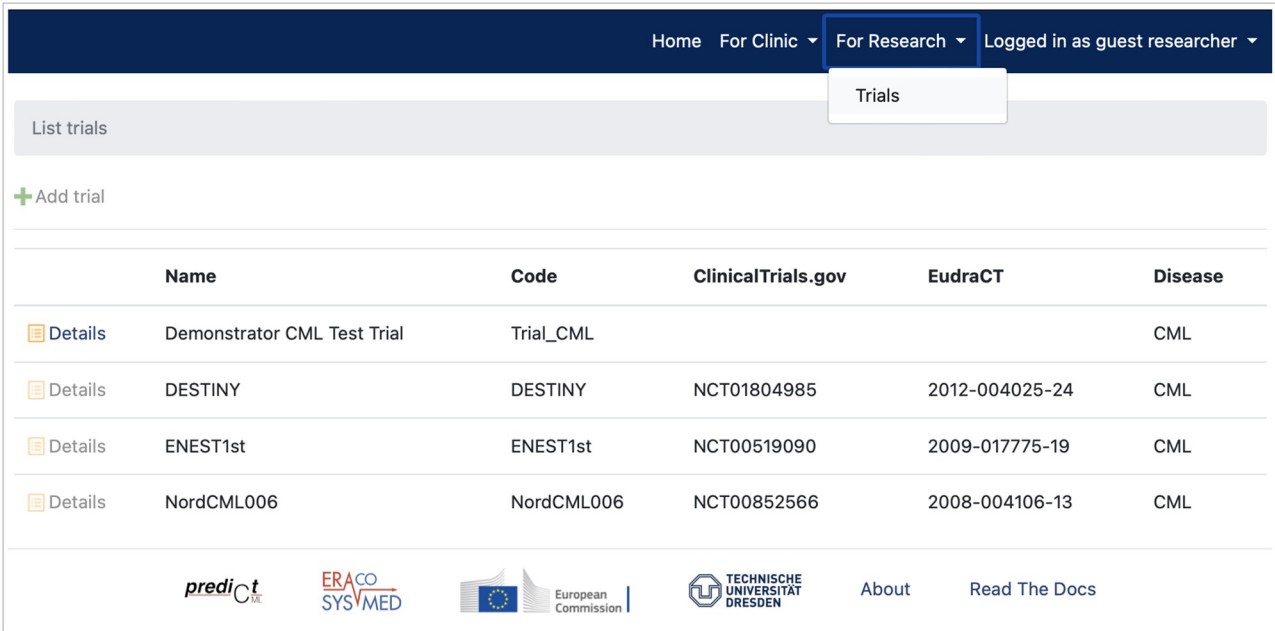

**Fig 5. Research view—List of collected trials.** Dependent on permission, the user can retrieve the detailed information of a trial, here the guest researcher is only able to retrieve the pseudonymized data and computer analyses of the "Demonstrator CML Test Trial".

to multiple users (in different contexts) for analyses processing (e.g., modelling/simulation) and how analytic results (e.g., mathematical model predictions) can be returned for direct application in clinical practice, ensuring data protection and ethical requirements.

As a result, a generic software framework (c.f. Fig 2) was designed that allows for seamless integration of computational applications into a common concept of Data Management in hospitals. Specifically, we demonstrate the integration and use of a particular application, denoted as "Integration of computational models and analytics". The application can also be regarded as a generic "container" of a variety of different computational sub-application/ algorithms.

In our particular example, we integrated two mathematical models, which provide different simulation-based predictions for disease and treatment dynamics in individual patients, in the context of TKI-treated CML. Our approach assumes that the data needed for these particular models are available in a (Research) Data Management System. In our specific examples, these are BCR-ABL/ABL time courses as molecular diagnostic data as well as TKI therapy information. We also assume that a pseudonymization service is established to provide and manage the particular patient-identifying data, application-specific consents and pseudonyms.

Our work continues earlier efforts to demonstrate how mathematical model predictions and analytics can be integrated into a specific clinical information system (CIS) [15]. In contrast to these previous studies, we are now presenting a stand-alone Clinical Decision Support System (CDSS) that is intrinsically connected to a (Research) Data Management System (c.f. Fig 1). This has the following advantages over the previous framework: 1) There is no limitation to the data collected in the dedicated CIS as it was in the previous framework. This means that any medical data required for computer simulation and analyses can be retrieved as long as it is stored in a clinical information system (e.g., laboratory systems, study databases). 2) Data quality is an enormously high priority. Therefore, data retrieved from the Research Data Management are cleansed before integration into the CDR. Suitable processes and methods to

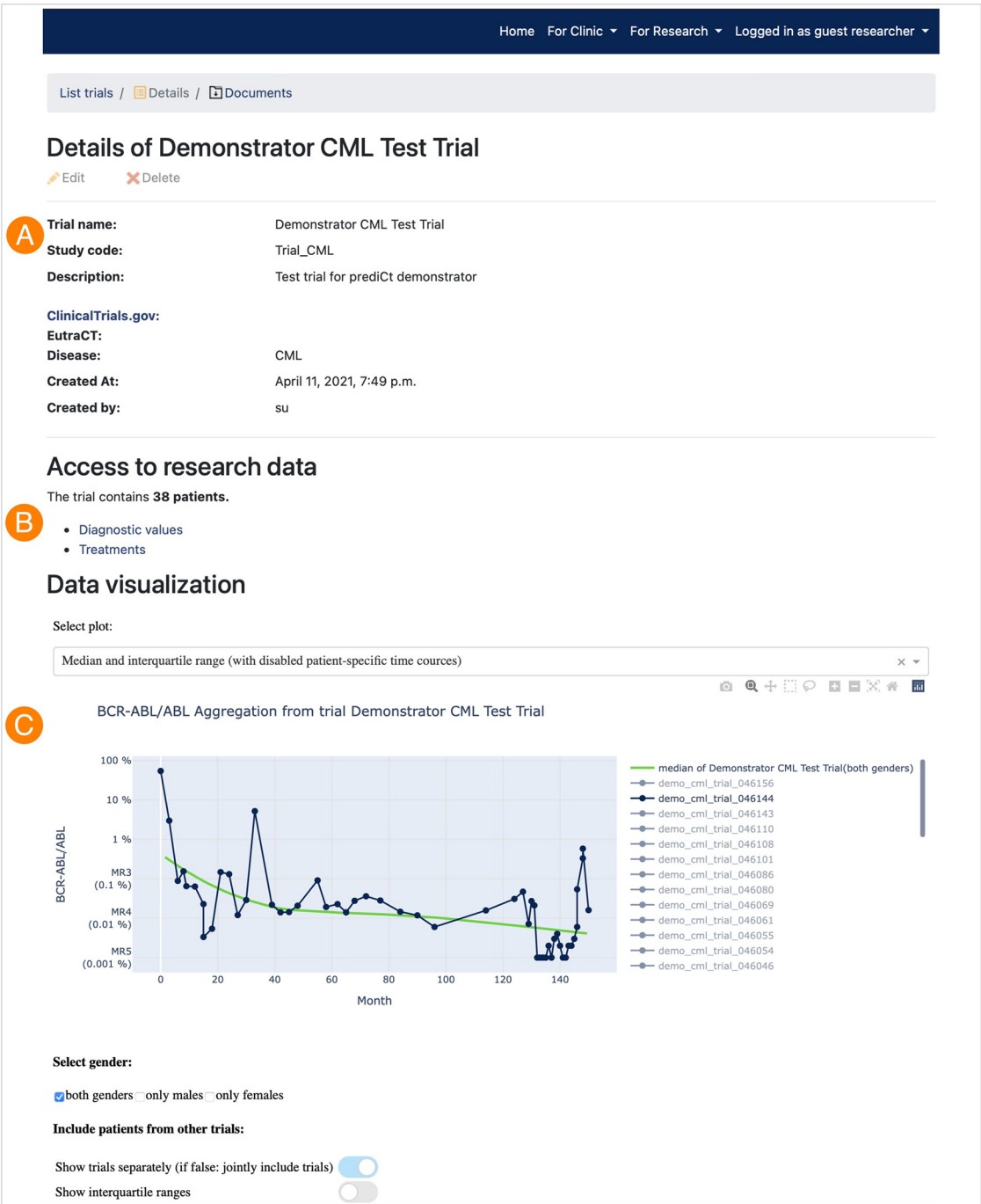

**Fig 6. Research view—Details of a trial.** This detailed view consists of general information (A), the ability to retrieve and download the current pseudonymized medical data (B) and the visualization of particular medical data and analysis on aggregated data (C).

achieve this goal are already established in a common hospital setting. 3) The integration of individual model predictions and analyses is independent of the technical capabilities of the used CIS, as the application is designed as a standalone application, unlike the previous framework.

Another important advancement of the presented framework (compared to [15]) is the integration of an independent Trusted Third Party/pseudonymization service as a core element of the research data management infrastructure. Like others [17,18,35], we illustrate how healthcare data can be used for scientific research in a manner that is compliant with data protection regulations. However, to our knowledge, this is the first example to illustrate that this service can also be used to provide patient-specific findings in a backwards manner to directly support clinical decision-making. As of now, we are only aware of two pseudonymisation services [26,36] while only one of them (the MOSAIC Tools) offers the necessary functionalities within our framework (i.e. identity, consent and pseudonym management). The advantages of other pseudonymization services should be addressed in future research projects.

The novel aspect of our current approach lies in the simultaneous provision and usage of health data in both clinical and research contexts (depending on the particular user context), which was not possible with the previous framework. The suggested structure is not limited to the outlined molecular time course data but can be used for a wide range of health data (e.g., from various CIS, laboratory systems, biobanks) to be integrated for joined analyses, simulations and visualizations. This simultaneous usage is particularly beneficial for clinical scientists in the treatment context since all relevant medical information is available from various primary data sources within a unified and dedicated framework.

Our general concept is also flexible in terms of the computational models to be integrated. This means that integrated models/algorithms can be flexible and as complex as necessary. Also, the presentation of the results in the clinic and research view can be designed as individually as the use case requires. It is, for example, also possible to implement a computer model in the Model and Simulation Server to artificially generate data that can be provided to the researcher as input for other applications, e.g., for the training of machine learning algorithms or simulation studies.

Future research should, therefore, investigate which model predictions and analyses are useful and what are the requirements for a user-friendly CDSS within a medical speciality (e.g., the integration within a CIS/EHR, the need for a comprehensive and understandable model description or the need for further and repeated consistency checks of the computer models). Furthermore, usability studies can be used to evaluate effectiveness and satisfaction and improve the layout and functionalities.

Standardized processes (e.g., CRISP-DM [37]) should be applied to support the process of computer model/application development, evaluation, deployment, and continuous improvement. Therefore, it should also be investigated how the software framework can be extended to support the implementation of such processes. In this context, we would also like to point out that, depending on the intended use of the application, compliance with regulations such as the MDR (Medical Device Regulation) or the In Vitro Diagnostic Regulation (IVDR) must also be ensured.

Moreover, given the extensive efforts regarding collaboration, knowledge sharing, and networking at the national [38,39] and international levels [40,41], it is equally important for future research to implement (and define as appropriate) common data models (e.g., using standards like FHIR (Fast Healthcare Interoperability Resources [42]), OMOP CDM (Observational Medical Outcomes Partnership Common Data Model [43]) to ensure interoperability and scalability of the solution.

While we extensively discussed the joined provision of both data and models in a clinical context, we did not elaborate on how the model predictions can actually be compared with future treatment outcomes. Such comparison is another step that iteratively contributes to the validation and continuous improvement (possibly automated by artificial intelligence) of the

models. Another useful aspect for further extension of the framework would be the integration of federated learning concepts within the provision of models.

In summary, this paper illustrates and describes how data analytic results, particularly in silico predictions, can be seamlessly integrated into workflows for simultaneous use in research and clinical care routines in a specific hospital setting. We believe that the presented software framework is an ideal basis for further, even commercial developments and the transfer into clinical practice.

## Supporting information

**S1 File. Walkthrough video of the demo server.**
(MP4)

## Acknowledgments

The Article Processing Charges (APC) were funded by the joint publication funds of the TU Dresden, including Carl Gustav Carus Faculty of Medicine, and the SLUB Dresden as well as the Open Access Publication Funding of the DFG.

## Author Contributions

**Conceptualization:** Katja Hoffmann, Ingmar Glauche, Ingo Roeder.

**Data curation:** Elena Karg, Andrea Gottschalk, Thomas Zerjatke.

**Funding acquisition:** Ingo Roeder.

**Methodology:** Katja Hoffmann, Anne Pelz, Elena Karg, Andrea Gottschalk, Thomas Zerjatke, Silvio Schuster, Ingmar Glauche.

**Resources:** Ingo Roeder.

**Software:** Katja Hoffmann.

**Supervision:** Ingmar Glauche, Ingo Roeder.

**Visualization:** Katja Hoffmann.

**Writing – original draft:** Katja Hoffmann, Ingmar Glauche, Ingo Roeder.

**Writing – review & editing:** Katja Hoffmann, Anne Pelz, Heiko Böhme, Ingmar Glauche, Ingo Roeder.

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
