## [Decision Letter · Decision Letter 0]

28 Dec 2022

PDIG-D-22-00296

Data integration between clinical research and patient care: a framework for context-depending data sharing and in silico predictions

PLOS Digital Health

Dear Dr. Hoffmann,

Thank you for submitting your manuscript to PLOS Digital Health. After careful consideration, we feel that it has merit but does not fully meet PLOS Digital Health's publication criteria as it currently stands. Therefore, we invite you to submit a revised version of the manuscript that addresses the points raised during the review process.

Please submit your revised manuscript within 60 days Feb 26 2023 11:59PM. If you will need more time than this to complete your revisions, please reply to this message or contact the journal office at digitalhealth@plos.org. Please include the following items when submitting your revised manuscript:

We look forward to receiving your revised manuscript.

Kind regards,

Ludwig Christian Giuseppe Hinske

Academic Editor

PLOS Digital Health

Journal Requirements:

1. We ask that a manuscript source file is provided at Revision. Please upload your manuscript file as a .doc, .docx, .rtf or .tex.

Additional Editor Comments (if provided):

Dear authors,

thank you very much for submission of your manuscript. As you can see from the reviewers' suggestions, all reviewers liked the manuscript. However, I would like you to address the raised points, especially to expand your analyses to better distinguish the current paper from previous work you have done in the field.

Looking forward to the revised manuscript,

 Christian Hinske

Reviewers' comments:

Reviewer's Responses to Questions

**Comments to the Author**

1. Does this manuscript meet PLOS Digital Health’s publication criteria? Is the manuscript technically sound, and do the data support the conclusions? The manuscript must describe methodologically and ethically rigorous research with conclusions that are appropriately drawn based on the data presented.

Reviewer #1: Yes

Reviewer #2: Partly

Reviewer #3: Yes

2. Has the statistical analysis been performed appropriately and rigorously?

Reviewer #1: N/A

Reviewer #2: N/A

Reviewer #3: N/A

3. Have the authors made all data underlying the findings in their manuscript fully available (please refer to the Data Availability Statement at the start of the manuscript PDF file)?

Reviewer #1: Yes

Reviewer #2: No

Reviewer #3: Yes

4. Is the manuscript presented in an intelligible fashion and written in standard English?

Reviewer #1: Yes

Reviewer #2: Yes

Reviewer #3: Yes

5. Review Comments to the Author

Reviewer #1: Thank you for the possibility to review this manuscript. The manuscript provides an interesting description of a software solution that integrates health data and computational analytics (e.g., model predictions, statistical evaluations, visualizations) into a clinical software solution which simultaneously supports both patient-specific healthcare decisions and research efforts. A link to the demonstrator is provided which is very helpful to the reader audience.

Abstract

• Insights from research transferred into clinical routine � implemented into clinical practice

• Software solution: it is much more that software, it is data standardisation, curation, analyses and implementation and you need (software) tools to support this. 

• To truly implement insight from research you need tools integrated with / within the EHR/CIS. I would expect some reference to this, also in the abstract

Introduction

• Please write CML in full the first time you use this abbreviation in the full text.

• It is not clear to me what you mean by “residual disease levels”

• Have you considered to compare your framework with general frameworks such as CRISP-DM or Gardner’s AI maturity model?

• Next to the mentioned 6 requirements I miss the continuous monitoring of the fit of the model in a changing clinical practice.

• The introduction is relatively long and includes both background and problem description as well as some theoretical framework on which the result is base. I would suggest to split this part into an introduction and a method section (which is position after the results section). Please end the introduction with a clear description of the aim of this study. In the method section you can somewhat better clarify the choices of your theoretical framework (which choices made by who and for what reason).

Results and some point for discussion

• Provide the link to the demonstrator earlier in the result section

• Have you considered to use syntaxtic data instead of pseudonimised data for the research application?

• In fig 2 it seems that data is used in the purper block at the bottom (to build models) is separated from the pseudonimisation while I believe that you use pseudonimised data for model development.

• Why is the pseudonimised data reidentified? I think you use pseudonimised data to develop a model, then apply this model on new cases in practice that can not be and donot need to be pseudonimised in the context of routine care. But the patients used to develop the model do not need to be deidentified, is not it?

• What does the abbreviation gICS mean?

• You wrote “This ensures that only data from patients who gave their written consent 217 to the respective use are finally provided” but patients do not have to give consent that their data is applied to a developed model in routine care is not this? Or is this legalisation specific in your country?

• How is uncertainty on the individual prediction modelled and visualised? Does the software solution provide any specific part to monitor the performance of the model and if needed recalibrate the model

• How is or can the demonstrater be integrated with the CIS/EHR?

• The topic of data pseudonimisation got a lot of attention but the also mentioned data quality aspects are nearly described. What does this solution do in that regard?

• The demonstrator presents results from the prediction model for some demo patients but it does not help the physician to understand how the model came to this prediction (which predictors contribute most to the outcome). This is know to be a very important ffeature ot get models implemented and used in practice.

Reviewer #2: The manuscript deals with an important question within the medical informatics community: providing useful recommendations to clinicians and researchers re-using clinical data, including data privacy protection technologies. The proposed framework is successfully tested and the results are reported in a visual and clear way. 

Major revision

In my opinion, the manuscript could address the research question in a more innovative manner, as the previous work (Reference n.15) already successfully demonstrates the integration of model results in clinical practice and suggests a framework for it. This new version of the framework includes new technologies and interesting innovations such as the TTP and the inclusion of the MOSAIC Tools, as well as updated models and dashboard. Prior publication, I would suggest a more thorough comparison of the two frameworks and an evaluation of the current one in terms of efficiency and satisfaction. 

According to the journal guidelines, the source code of the system and the models must be published. 

Minor revision

L37„and a research perspective focusing on the exploration of aggregated, but pseudonymized data.“

Aggregated but pseudonymized is contradicting.

L76 „we need to ask how health data“ 

I would rephrase this.

L113 „loading into a data warehouse“ 

„Data warehouse“ does not represent all of the end targets of an ETL process. 

L117 „This ensures that the pseudonymized medical data do not allow any conclusions about a patient's identity.“

This statement could be refuted.

L135 „data protection laws“

What laws exactly?

L156 „standards for data security and pseudonymisation“

What standards?

Reviewer #3: Hoffmann et al. describe a relatively comprehensive and practical proposal for both sharing pseudonymized patient record data derived from the EHR systems for predictive modelling, and returning the model data via re-identification for individual patient use. 

The manuscript text is well written, concise, and sufficient in detail. Figures would benefit from editing by a medical graphics designer/artist.

Specific questions

1. Is the proposed system compatible with common cloud infrastructures (e.g. Microsoft Azure) used for EHRs for most hospitals.

2. Has the solution been installed and validated in production use in a real hospital environment? Any experience on e.g. Epic integration?

3. The authors describe the solution as “generic”, yet the use cases presented are the same from their previous work – CML outcome modeling. As CML is an exceptionally “simple” cancer with only one marker (blood BCR::ABL1-transcript level) sufficient for patient follow-up. How does the “clinical view” handle longitudinal visualization of more complex diseases, like breast cancer, with multiple markers (e.g. ctDNA) and other outcome measures (radiology, pathology) routinely used for follow-up and outcome modeling?

4. Would be good to mention that implementation of a common data model for the clinical data (e.g. FHIR, OMOP) would ensure scalability of the solution to other environments and data owners

5. Can the solution utilize data from multiple data owners in a federated way (no central data repository; no transfer of primary patient data)?

6. Will the solution likely be EHDS-compliant?

6. PLOS authors have the option to publish the peer review history of their article (what does this mean?). If published, this will include your full peer review and any attached files.

**Do you want your identity to be public for this peer review?** For information about this choice, including consent withdrawal, please see our Privacy Policy.

Reviewer #1: No

Reviewer #2: No

Reviewer #3: No

---

## [Decision Letter · Decision Letter 1]

30 Mar 2023

Data integration between clinical research and patient care: a framework for context-depending data sharing and in silico predictions

PDIG-D-22-00296R1

Dear Mrs Hoffmann,

We are pleased to inform you that your manuscript 'Data integration between clinical research and patient care: a framework for context-depending data sharing and in silico predictions' has been provisionally accepted for publication in PLOS Digital Health.

Best regards,

Ludwig Christian Giuseppe Hinske

Academic Editor

PLOS Digital Health

Reviewer Comments (if any, and for reference):

Reviewer's Responses to Questions

**Comments to the Author**

1. If the authors have adequately addressed your comments raised in a previous round of review and you feel that this manuscript is now acceptable for publication, you may indicate that here to bypass the “Comments to the Author” section, enter your conflict of interest statement in the “Confidential to Editor” section, and submit your "Accept" recommendation.

Reviewer #1: All comments have been addressed

Reviewer #2: All comments have been addressed

Reviewer #3: All comments have been addressed

2. Does this manuscript meet PLOS Digital Health’s publication criteria? Is the manuscript technically sound, and do the data support the conclusions? The manuscript must describe methodologically and ethically rigorous research with conclusions that are appropriately drawn based on the data presented.

Reviewer #1: Yes

Reviewer #2: Yes

Reviewer #3: Yes

3. Has the statistical analysis been performed appropriately and rigorously?

Reviewer #1: I don't know

Reviewer #2: I don't know

Reviewer #3: Yes

4. Have the authors made all data underlying the findings in their manuscript fully available (please refer to the Data Availability Statement at the start of the manuscript PDF file)?

Reviewer #1: Yes

Reviewer #2: Yes

Reviewer #3: Yes

5. Is the manuscript presented in an intelligible fashion and written in standard English?

Reviewer #1: Yes

Reviewer #2: Yes

Reviewer #3: Yes

6. Review Comments to the Author

Reviewer #1: I am fully satisfied by the answers the authors gave to my remarks

Reviewer #2: (No Response)

Reviewer #3: Thank you for carefully revising the manuscricpt according to the reviewer's comments.

7. PLOS authors have the option to publish the peer review history of their article (what does this mean?). If published, this will include your full peer review and any attached files.

**Do you want your identity to be public for this peer review?** For information about this choice, including consent withdrawal, please see our Privacy Policy.

Reviewer #1: No

Reviewer #2: No

Reviewer #3: **Yes: **Kimmo Porkka
